# DEGRADATION-AWARE UNFOLDING KNOWLEDGE-ASSIST TRANSFORMER FOR SPECTRAL COMPRESSIVE IMAGING

## ABSTRACT

Snapshot compressive spectral imaging offers the capability to effectively capture three-dimensional spatial-spectral images through a single-shot two-dimensional measurement, showcasing its significant potential for spectral data acquisition. However, the challenge of accurately reconstructing 3D spectral signals from 2D measurements persists, particularly when it comes to preserving fine-grained details like textures, which is caused by the lack of high-fidelity clean image information in the input compressed measurements. In this paper, we introduce a two-phase training strategy embedding high-quality knowledge prior in a deep unfolding framework, aiming at reconstructing high-fidelity spectral signals. Our experimental results on synthetic benchmarks and real-world datasets demonstrate the notably enhanced accuracy of our proposed method, both in spatial and spectral dimensions. Code and pre-trained models will be released.

## 1 INTRODUCTION

Hyperspectral images (HSIs) contain multiple spectral bands with more abundant spectral signatures than normal RGB images, widely applied in image classification (Maggiori et al., 2017; Li et al., 2019), object detection (Li et al., 2020; Rao et al., 2022), tracking (Van Nguyen et al., 2010; Uzkent et al., 2017), medical imaging (Lu & Fei, 2014; ul Rehman & Qureshi, 2021), remote sensing (Goetz et al., 1985; Lu et al., 2020), etc. To collect HSI, conventional imaging systems use spectrometers to scan the scenes along the spectral or spatial dimension, which is time-consuming and limited to static objects. Recently, snapshot compressive imaging (SCI) systems (Du et al., 2009; Llull et al., 2013; Cao et al., 2016; Luo et al., 2023) have obtained much attention to capture HSIs, among which the coded aperture snapshot spectral imaging (CASSI) Wagadarikar et al. (2008b); Meng et al. (2020c) stands out with impressive performance and efficiency. CASSI modulates the HSI signal at

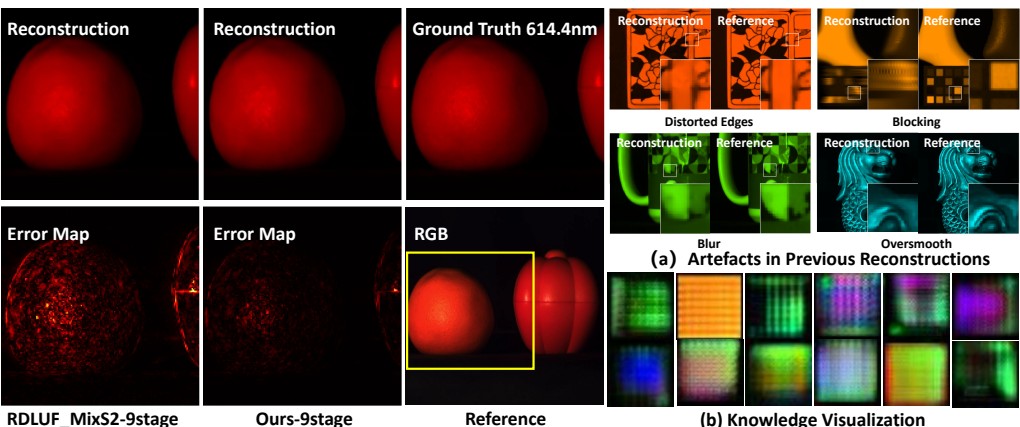

Figure 1: Comparisons with the previous reconstruction methods, and the visualization of learned knowledge. The error map shows the absolute error between the reconstructed spectral image and the ground truth, where brighter areas indicate larger error values. The zoomed parts in subfigures in (a) show several typical artefacts we found in previous reconstruction methods, including distorted edges, blocking, blur and oversmooth. The visualization of the learned knowledge in (b) further illustrates the learned patterns facilitate image fidelity.

different wavelengths and mixes all modulated spectra to generate a 2D compressed measurement. Subsequently, reconstructing the 3D HSI cube from the 2D compressive measurements poses a fundamental challenge for the CASSI system.

From traditional model-based methods Bioucas-Dias & Figueiredo (2007); Beck & Teboulle (2009); Yuan (2016) to widely used learning-based methodsCharles et al. (2011); Miao et al. (2019); Meng et al. (2020b), CASSI reconstruction methods have been developed for years to recover high-quality HSI signals. Recent studies adopt deep unfolding Wang et al. (2022); Cai et al. (2022c); Meng et al. (2023); Dong et al. (2023) framework with a multi-stage network to map the measurement into the HSI cube. These deep unfolding networks are intuitively interpretable by explicitly characterizing the image priors and the imaging system. Besides, these methods also enjoy the power of learning-based methods (deep denoisers or prior networks) and thus have great potential. Furthermore, with the help of the representation capabilities on the long-range dependency of the Transformer Vaswani et al. (2017), the deep denoisers of unfolding networks can explore the spatial and spectral correlations Hu et al. (2022); Cai et al. (2022b); Dong et al. (2023).

Previous reconstruction networks aim to recover the clean image from compressed measurement, which encourages learning the local and non-local similarity in spatial and spectral dimensions from measurement. However, there remains an intrinsic issue implied in the ill-posed CASSI inverse problem, where the compressed measurement is severely degraded due to physical modulation, spectral compression, and unpredictable system noise. Thus, the severe degradation mismatch leads to insufficient authenticity and limited fidelity in the reconstructed images, such as distorted edges, blocky patterns, blurring, and other artifacts (as shown in Fig. 1).

To solve the problem, in this paper, we leverage high-fidelity knowledge, *i.e.* discrete prototypes learned from the uncompressed images, and explicit degradation information into the proposed deep unfolding framework for spectral compressive image reconstruction. Specifically, we train a lightweight counselor model by constructing a vector quantized variational autoencoder (VQ-VAE) to learn high-fidelity knowledge from the ground-truth image space. Then, we proposed a U-net bridge denoiser equipped with the mask-aware knowledge-assist attention (MKA) mechanism to incorporate multi-layer degradation information and high-fidelity prior knowledge in an elegant and principal manner. Notably, our unfolding framework can absorb external prior knowledge from the counselor model and the cascade degradation information to boost its reconstruction performance. The main contributions we have made in this paper can be summarized as follows:

*i)* We propose a degradation-aware knowledge-assist deep unfolding framework using vector-quantized high-fidelity HSI knowledge to guide the CASSI reconstruction.

*ii)* We propose a U-net bridge denoiser that integrates high-fidelity HSI knowledge and cascade degradation information to guide the CASSI reconstruction, which is learned with a VQ-VAE-based counselor network and a sophisticated mask cross encoder, respectively.

*iii)* Extensive experiments on the synthetic benchmark and real dataset demonstrate the superior accuracy of our proposed method both in spatial and spectral dimensions.

## 2 RELATED WORK

### 2.1 HYPERSPECTRAL IMAGE RECONSTRUCTION

Traditional model-based methods Wang et al. (2017); Zhang et al. (2019); Wagadarikar et al. (2008a) are mainly based on hand-crafted image priors, such as total variation Yuan (2016), sparsity Kittle et al. (2010), low-rank Liu et al. (2019), etc. While these methods have demonstrated theoretical properties and interpretability, they require manual parameter tuning with low reconstruction speed and suffer from limited representation capacity and poor generalization ability. Recently, deep learning methods have been used to solve the inverse problem of spectral SCI. The first popular branch is Plug-and-play (PnP) algorithms Chan et al. (2017); Ryu et al. (2019); Yuan et al. (2020; 2021), which integrate pre-trained denoising networks into traditional model-based methods to solve the HSI reconstruction problem, but suffering from fixed nature of pre-trained denoiser and limited performance. The second branch of deep learning methods follows the End-to-end (E2E) training manner, which usually employ a powerful network, such as convolutional neural network (CNN) Cheng et al. (2022); Lu et al. (2020); Hu et al. (2022) and Transformers Cai et al. (2022b;a), to learn the mapping function from measurements to desired HSIs. However, they learn a brute-force mapping

ignoring the working principles of CASSI systems, thereby lacking interpretability and flexibility for various hardware systems. To overcome these issues, Deep unfolding methods Meng et al. (2020a); Ma et al. (2019); Cai et al. (2022c); Dong et al. (2023) take advantage of both model-based methods and deep learning-based methods, which transfer conventional iterative optimization algorithms into the multi-stage network. Each stage typically involves a linear projection and a single-stage network that learns the underlying denoiser prior. Deep unfolding methods offer intuitive interpretability by explicitly modeling image priors and the system. Moreover, they benefit from the power of deep learning, presenting significant potential that remains partially untapped.

## 2.2 PROTOTYPE LEARNING FOR SCI

The key idea of prototype learning for SCI is to exploit the non-local similarity as prototypes across images. In the SCI problem considered here, the non-local similarity can also be searched across different frames. In previous methods, sparse representation with learned dictionaries has demonstrated its superiority in SCI reconstruction, such as sparse dictionary (usually over-complete) learning Aharon et al. (2006), Bayesian dictionary learning Yuan et al. (2015), group-based sparse coding Liu et al. (2019). However, these methods usually require an iterative optimization to learn the dictionaries and sparse coding or pay a price for running time in patch matching, suffering from high computational costs. With the development of VQVAE Van Den Oord et al. (2017), the first method introduced a highly compressed codebook, is realizable to learn the representative prototypes with a vector-quantized Autoencoder model. Unlike the large hand-crafted dictionary Jo & Kim (2021), the learnable codebook automatically learns optimal elements and provides superior efficiency and expressiveness, circumventing the laborious dictionary design Zhou et al. (2022). Inspired by prototype learning, this paper investigates a discrete proxy space for SCI reconstruction. Different from existing methods, we exploit the discrete unpolluted iconic prototypes with a counselor model from the ground truth video as external knowledge prior and aim to guide the reconstruction. Such designs allow our method to take full advantage of the knowledge so that it does not solely depend on the degraded compressed information, tightly fitting the spectral compressive imaging task and significantly enhancing the robustness of face restoration.

## 3 PROPOSED MODEL

### 3.1 PROBLEM FORMULATION

The CASSI system aims to modulate different wavelengths in the spectral data cube and then integrate them into a 2D imaging sensor. The mathematical model of the single-disperser CASSI (SD-CASSI) Wagadarikar et al. (2008a) sensing process is illustrated as follows. As shown in the Fig. 2 (a), the original HSI data, denoted as $\boldsymbol{X} \in \mathbb{R}^{W \times H \times B}$, is modulated by the fixed physical mask $\boldsymbol{M} \in \mathbb{R}^{W \times H}$, where $W$, $H$, and $B$ denote the width, height, and the number of spectral channels, respectively. The coded HSI data cube is represented as $\mathbf{X}'(:,:,b) = \mathbf{X}(:,:,b) \odot \mathbf{M}, b = 1, 2, \ldots, B$, where $\odot$ represents the element-wise multiplication. After light propagating through the disperser, each channel of $\mathbf{X}'$ is shifted along the $\boldsymbol{H}$-axis. The shifted data cube is denoted as $\mathbf{X}'' \in \mathbb{R}^{W \times \tilde{H} \times B}$, where $\tilde{H} = H + d_B$. Here, $d_B$ represents the shifted distance of the $B - th$ wavelength. This process can be formulated as modulating the shifted version $\tilde{\mathbf{X}} \in \mathbb{R}^{W \times \tilde{H} \times B}$ with a shifted mask $\tilde{\mathbf{M}} \in \mathbb{R}^{W \times \tilde{H} \times B}$, where $\tilde{\boldsymbol{M}}(i, j, b) = M(w, h + d_B)$. At last, the imaging sensor captures the shifted image into a 2D measurement $\mathbf{Y}$, calculated as

$$\mathbf{Y} = \sum_{b=1}^{B} \tilde{\mathbf{X}}(:,:,b) \odot \tilde{\mathbf{M}}(:,:,b) + \mathbf{N}, \tag{1}$$

where $\mathbf{N} \in \mathbb{R}^{W \times \tilde{H}}$ denotes the measurement noise. By vectorizing the spectral data cube and measurement, that is $x = \text{vec}(\tilde{\mathbf{X}}) \in \mathbb{R}^{W \tilde{H} B}$ and $y = \text{vec}(\mathbf{Y}) \in \mathbb{R}^{W \tilde{H}}$, this model can be formulated as

$$y = \mathbf{A}x + n, \tag{2}$$

where $\mathbf{A} \in \mathbb{R}^{W \tilde{H} \times W \tilde{H} B}$ denotes the sensing matrix (coded aperture) which is a concatenation of diagonal matrices, that is $\mathbf{A} = [\mathbf{D}_1, \ldots, \mathbf{D}_B]$, where $\mathbf{D}_b = Diag(vec(\tilde{\mathbf{M}}(:,:,b)))$ is the diagonal matrix with $vec(\tilde{\mathbf{M}}(:,:,b))$ as the diagonal elements. The sensing matrix $\mathbf{A}$ is a sparse matrix and $\boldsymbol{A}\boldsymbol{A}^\top$ is a diagonal matrix Jalali & Yuan (2019). In this paper, we focus on the ill-posed HSI restoration problem, recovering the high-quality image $x$ from the compressed measurement $y$.

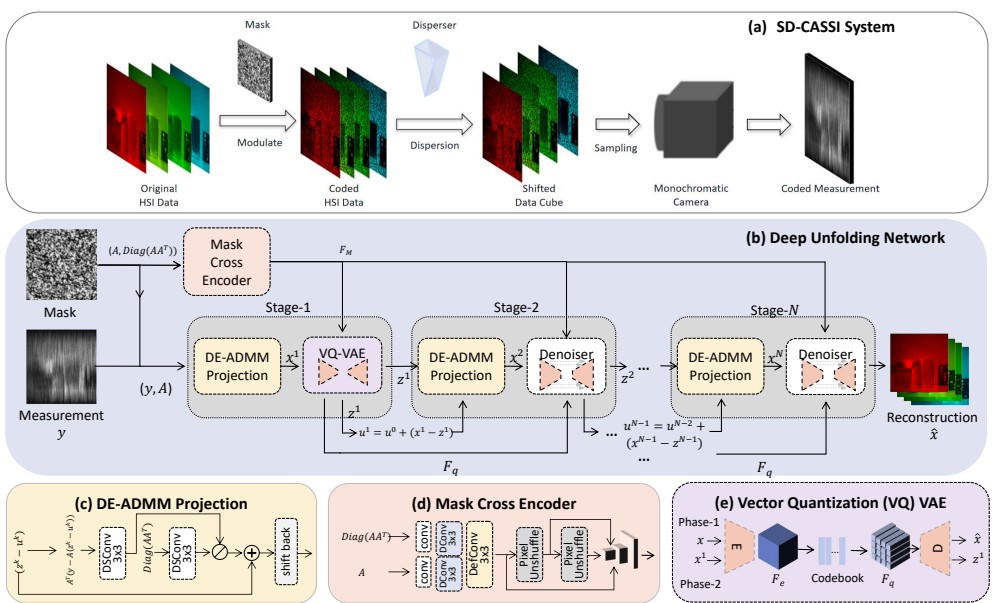

Figure 2: (a) The SD-CASSI forward process. HSI data cube is coded and compressed on a 2D sensor plane. (b) Our unfolding framework. (c) DE-ADMM projection. (d) Mask cross encoder. (e) Vector quantized Variational AutoEncoder (VQ-VAE). In phase 1, only VQ-VAE is trained in a self-supervised manner, both input and output are clean HSIs. In phase 2, VQ-VAE is embedded into the unfolding framework and updates its encoder parameter with other unfolding structures. Mask $A$ and measurement $y$ are input in this phase.

## 3.2 THE UNFOLDING ADMM FRAMEWORK

HSI reconstruction problem in Eq. 2 can be typically solved by convex optimization through the following objective:

$$\hat{\boldsymbol{x}} = \arg\min_{\boldsymbol{x}} \frac{1}{2}\|\boldsymbol{y} - \boldsymbol{A}\boldsymbol{x}\|_2^2 + \lambda R(\boldsymbol{x}), \tag{3}$$

where $\lambda$ is a noise-balancing factor. The first term guarantees that the solution $\hat{\boldsymbol{x}}$ meets the observation, and the second term $R(\boldsymbol{x})$ refers to the image regularization.

ADMM (Alternating Direction Method of Multipliers) Boyd et al. (2011) breaks down the original optimization problem into multiple sub-problems and updates them alternately, leading to excellent convergence, error reduction, reduced computational and storage complexity. Here, we choose ADMM as our optimization framework Zheng et al. (2021). Therefore, the problem in Eq. 3.2 can be written as:

$$\boldsymbol{x}^{k+1} = \arg\min_{\boldsymbol{x}} \frac{1}{2}\|\boldsymbol{A}\boldsymbol{x} - \boldsymbol{y}\|_2^2 + \frac{\rho}{2}\left\|\boldsymbol{x} - \left(\boldsymbol{z}^k - \boldsymbol{u}^k\right)\right\|_2^2, \tag{4}$$

$$\boldsymbol{z}^{k+1} = \arg\min_{\boldsymbol{z}} \lambda R(\boldsymbol{z}) + \frac{\rho}{2}\left\|\boldsymbol{z} - \left(\boldsymbol{x}^{k+1} + \boldsymbol{u}^k\right)\right\|_2^2, \tag{5}$$

$$\boldsymbol{u}^{k+1} = \boldsymbol{u}^k + \left(\boldsymbol{x}^{k+1} - \boldsymbol{z}^{k+1}\right), \tag{6}$$

where $\boldsymbol{z}$ is an auxiliary variable, $\boldsymbol{u}$ is the multiplier, $\rho$ is a penalty factor, and $k$ is the index of iterations. Recalling the proximal operator Parikh et al. (2014), defined as $\text{prox}_g(\boldsymbol{v}) = \arg\min g(\boldsymbol{x}) + \frac{1}{2}\|\boldsymbol{x} - \boldsymbol{v}\|_2^2$, Equ. 5 is the Euclidean projection with a closed-form solution, i.e., $\boldsymbol{x}^{k+1} = \left(\boldsymbol{A}^\top\boldsymbol{A} + \rho\boldsymbol{I}\right)^{-1} \cdot \left[\boldsymbol{A}^\top\boldsymbol{y} + \rho\left(\boldsymbol{z}^k - \boldsymbol{u}^k\right)\right]$. Equ. 6 can be viewed as a denoiser $\mathscr{D}$. Furthermore, recalling that $AA^\top$ is a diagonal matrix for image-plane coding, $\left(\boldsymbol{A}^\top\boldsymbol{A} + \rho\boldsymbol{I}\right)^{-1}$ can be calculated efficiently using the matrix inversion lemma (Woodbury matrix identity) as $\left(\boldsymbol{A}^\top\boldsymbol{A} + \rho\boldsymbol{I}\right)^{-1} = \rho^{-1}\boldsymbol{I} - \rho^{-1}\boldsymbol{A}^\top\left(\boldsymbol{I} + \rho\boldsymbol{A}\boldsymbol{A}^\top\right)^{-1}\boldsymbol{A}\rho^{-1}$. Then the Euclidean projection can be simplified and the

final solution is

$$x^{k+1} = \left(z^k - u^k\right) + A^\top \left[y - A\left(z^k - u^k\right)\right] \oslash \left[\text{Diag}\left(AA^\top\right) + \rho\right], \tag{7}$$

$$z^{k+1} = \mathscr{D}_k\left(x^{k+1} + u^k\right), \tag{8}$$

$$u^{k+1} = u^k + \left(x^{k+1} - z^{k+1}\right), \tag{9}$$

where $\text{Diag}(\cdot)$ extracts the diagonal elements of the ensued matrix, $\oslash$ denotes the Hadamard division, and $\mathscr{D}_k$ is the denoiser of the $k-th$ stage. Here, the noise penalty factor $\rho$ is tuned to match the measurement noise. Considering the projection step Equ. 7, assisted by deep learning, we can correct this step as follows:

$$x^{k+1} = z_u^k + DE(A^\top(y - Az_u^k))DE\left(\text{Diag}\left(AA^\top\right)\right), \tag{10}$$

where $z_u^k = \left(z^k - u^k\right)$, and $DE(\cdot)$ denotes the deep learning enhancement in ADMM, implemented by a neural network consisting of a depthwise separable convolution and a GELU Hendrycks & Gimpel (2016) operation. This enhancement aims to use deep linear and non-linear operations, as illustrated in Fig. 2 (c), to further correct the gradient descent direction in ADMM and estimate the influence of noise in the diagonal matrix. Thus, we name this step Deep Enhanced ADMM (DE-ADMM) Euclidean Projection. The overall unfolding framework is shown in Fig. 2 (b), mask $A$ and measurement $y$ are inputs of the network, following the Equ. 7,8 and 9, we acquire the first stage output $z^1$ and residue $u^1$. The first stage denoiser is a pre-trained VQ-VAE with some fixed modules. The intermediate variable $\mathbf{F}_q$ will be used as prior knowledge in the later stages. Other stage denoisers are U-shape networks, receiving cross space mask features and prior knowledge. Therefore, we will next introduce how to learn the cross-space mask features and high-fidelity knowledge priors, and then propose a novel scheme to integrate them to facilitate denoising and reconstruction.

### 3.3 MASK-AWARE HIGH-FIDELITY KNOWLEDGE LEARNING

#### 3.3.1 MASK CROSS ENCODER

The mask is a crucial component in SCI reconstruction as it provides the degradation details in SCI measurement. However, establishing the relationship between the degradation information and the mask in compressed measurements can be challenging. To address this, we propose a Mask Cross Encoder (MCE) that extracts the Cross-Space Mask Feature from two Euclidean spaces: the observation and compressive domains. As shown in Fig. 2 (d), the mask ($\mathbf{A}$) and compressed mask ($Diag(\mathbf{AA}^\top)$) are fed into a dual-path network respectively, and then two kinds of information $F_{\mathbf{AA}^\top} = Conv(Sigmoid(Diag(\mathbf{AA}^\top)))$ and $F_{\mathbf{A}} = Conv(Sigmoid(\mathbf{A}))$ are concatenated and fused as:

$$F_{\text{fusion}} \in \mathbb{R}^{H \times W \times C} = DefConv(DConv(Concat(F_{\mathbf{A}}, F_{\mathbf{AA}^\top})))$$

where $DConv$ refers to the dilated convolution network, which effectively increases the receptive field while minimizing the number of parameters. $DefConv$ denotes the deformable convolution, which learns offsets based on object shapes within images, allowing for the extraction of more intricate information compared to vanilla convolution. In this way, the mask-aware features across two Euclidean spaces are fused and aligned to guide the SCI reconstruction.

Then we use the pixel unshuffle (PU) operations to obtain the multi-level mask-aware representations by zooming feature size. Each PU operation rearranges elements, transforming a tensor of shape $H \times W \times C$ to $\frac{H}{2} \times \frac{W}{2} \times 4C$. As shown in Fig. 2 (d), two PU layers are employed, and the outputs contain three multi-scale mask-aware features, stated as:

$$F_M^i = \{ \begin{matrix} F_{\text{fusion}} \text{ if } i = 1 \\ PU(F_M^{i-1}) \text{ if } i > 1 \end{matrix}, F_M^i \in \mathbb{R}^{\frac{H}{2^{i-1}} \times \frac{W}{2^{i-1}} \times 4^{i-1}C}, i = 1, 2, 3.$$

To this end, MCE encodes mask features into hierarchical representations and will facilitate SCI reconstruction by integrating the degradation information.

#### 3.3.2 VECTOR QUANTIZATION FOR HIGH-FIDELITY KNOWLEDGE LEARNING

In previous SCI reconstruction methods, the input typically consists of compressed measurements and masks that lack local textures and details. This limited information results in the lack of high-fidelity natural image prior knowledge for the reconstruction. Thus, we aim to solve this problem using discrete representation to learn high-fidelity knowledge from the uncompressed domain.

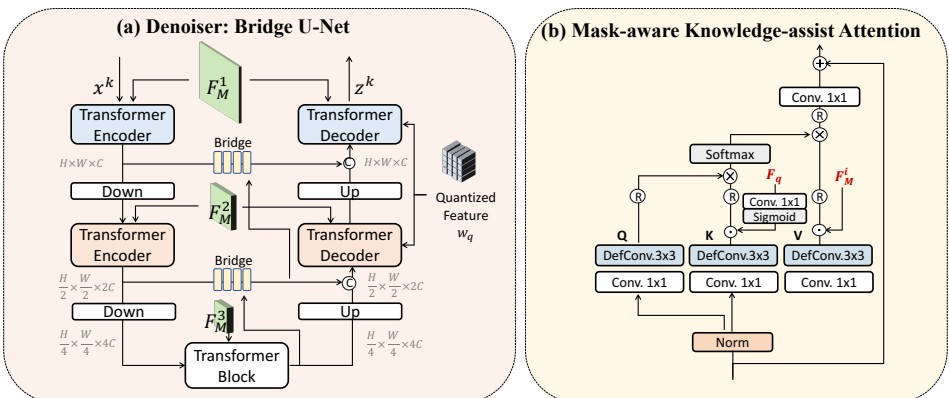

Figure 3: (a) The bridge U-Net. Each bridge module is composed of several convolution layers. The mask features obtained by the mask cross encoder will be sent into each encoder and decoder for cross-space reconstruction. (b) The Mask-aware Knowledge-assist Attention module (MKA), where the mask features $F_M^i$ and prior knowledge $F_q$ are integrated through cross attention.

Specifically, we design a two-phase vector quantized variational autoencoder to learn and integrate the discrete unpolluted knowledge.

**Phase 1:** We aim to learn the prior high-fidelity knowledge by a lightweight counselor model in a self-supervised manner, which encodes clean HSI data $x$ into $\mathbf{F}_e$ by the encoder $E$, then reconstructs $\hat{x}$ from $\mathbf{F}_q$ by the decoder $D$. Specifically, as shown in Fig. 2 (e), we first encode the $x \in \mathbb{R}^{W \times H \times B}$ into latent feature $\mathbf{F}_e \in \mathbb{R}^{m \times n \times d}$ output by the ADMM projection. Then, we quantize the continual feature $\mathbf{F}_e$ as $\mathbf{F}_q \in \mathbb{R}^{m \times n \times d}$ by replacing the features of each position with its nearest prototypes in the codebook $\mathscr{C} = \left\{ c_k \in \mathbb{R}^d \right\}_{k=0}^N$ along spatial dimension, formulated as:

$$\mathbf{F}_q^{(i,j)} = \arg\min_{c_k \in \mathscr{C}} \left\| \mathbf{F}_e^{(i,j)} - c_k \right\|_2 .$$

Then the quantized feature $\mathbf{F}_q$ is decoded into reconstruction $\hat{x}$ through decoder $D$. The training objective for optimizing VQ-VAE and codebook can be formulated as:

$$\mathscr{L}_{\text{codebook}} = \left\| x - \hat{x} \right\|_1 + \left\| \text{sg} \left( \mathbf{F}_e \right) - \mathbf{F}_q \right\|_2^2 + \beta \left\| \mathbf{F}_e - \text{sg} \left( \mathbf{F}_q \right) \right\|_2^2 ,$$

where $\text{sg}(\cdot)$ stands for the stop-gradient operator and $\beta$ is a trade-off weight. The first term of the objective aims to measure the reconstruction loss and the rest terms aim to regularize the high-fidelity prototypes in codebooks. The encoder-decoder network is implemented by an elegant U-Net, detailed in Section 3.4.1. The counselor model is inserted into the first stage of the unfolding framework as the first denoiser and shares the quantized $\mathbf{F}_q$ and framework with denoisers in the later stages.

**Phase 2:** At the previous phase, the high-fidelity prototypes learned in a self-supervised manner are reckoned as prior knowledge in the successive unfolding stages to reconstruct the high-quality HSI data. Fixing the decoder and codebook, we aim to leverage knowledge for training an unfolding network to reconstruct HSIs from measurement. In the second phase, the codebook and decoder are fixed, while the encoder adapts its parameters to encode $x^l$, which can be recognized as the noisy HSI. Thus, the training process in this phase is relatively easier compared to directly inputting the measurement. Notably, the training objective in the second phase only contains the reconstruction loss for denoiser updating.

## 3.4 Mask-aware knowledge-assist Denoising

### 3.4.1 Overview of the Denoiser: Bridge U-Net

Obtained the mask-aware features and high-fidelity prior knowledge, how to make good use of them remains unsolved. Therefore, we propose a U-Net-shaped framework for denoising with information filtering bridges between the hierarchical symmetric encoders and decoders. Instead of using a direct

skip connection, we implement a bridge module including several convolution layers to aggregate information from each spatial level and filter unimportant information. Furthermore, the features from the encoder and the filtered features of a higher-level bridge are concatenated and fused as a 'clean' residual information for decoding. As shown in Figure 3 (a), the multi-level mask features and quantized features are integrated for better reconstruction, implemented with the proposed mask-aware knowledge-assist attention module in transformer encoder and decoders.

### 3.4.2 LEVERAGING THE MASK AND HIGH-FIDELITY INFORMATION INTO TRANSFORMER

In the Transformer block, we introduce the multi-scale mask features $F_M^i$ and quantized high-fidelity HSI feature $\mathbf{F}_q$ into the vanilla multi-head attention, named Mask-aware Knowledge-assist Attention module (MKA). As shown in the Fig. 3 (b), the MKA encodes input $\mathbf{U}_i$ by $1 \times 1$ convolutions and deformable convolutions into query ($\mathbf{Q}$), key ($\mathbf{K}$) and value ($\mathbf{V}$) (with the same size as $\mathbf{U}_i$). Here, we implement a cross attention, *i.e.* modulate $\mathbf{K}$ with selected high-fidelity HSI feature $Sigmoid(Conv(\mathbf{F}_q))$ and modulate $\mathbf{V}$ with mask feature $F_M^i$ while keep $\mathbf{Q}$ unchanged. The Sigmoid and convolution transform $\mathbf{F}_q$ into a proper latent space to interact with $\mathbf{K}$. The MKA can be formulated as:

$$\mathbf{MKA}(\mathbf{U}_i) = W_{c1}\mathbf{V} \odot Softmax(\mathbf{K} \odot \mathbf{Q}/\alpha) \tag{11}$$

$$\mathbf{Q} = \mathbf{W}_{c2}^Q(\mathbf{W}_d^Q Norm(\mathbf{U}_i) + \boldsymbol{b}_d^Q), \tag{12}$$

$$\mathbf{K} = \mathbf{W}_{c2}^K(\mathbf{W}_d^K Norm(\mathbf{U}_i) + \boldsymbol{b}_d^K) \odot Sigmoid(\mathbf{W}_d^{F_q}\mathbf{F}_q), \tag{13}$$

$$\mathbf{V} = \mathbf{W}_{c2}^V(\mathbf{W}_d^V Norm(\mathbf{U}_i) + \boldsymbol{b}_d^V) \odot \mathbf{F}_M^i, \tag{14}$$

where $\mathbf{W}_{c1}$ and $\mathbf{W}_{c2}$ represent weights of bias-free convolution, $\mathbf{W}_d, \boldsymbol{b}_d$ are weight and bias of the deformable convolution, $F_M^i, i = 1, 2, 3$ denotes the mask feature of different spatial levels, and $F_q$ is the quantized feature which refers to the learned prior knowledge. Leveraging mask features enables the encoder and decoder to make the most use of degradation information at each spatial level and align the position of the mask-relative noise on $x^k$. The high-fidelity knowledge learned from the uncompressed images provides guidance for high-quality reconstruction, as it preserves detailed GT features. In summary, the MKA module helps to focus on relevant information instead of falling into the noisy HSI data itself while considering the information of mask and prior HSI knowledge, investigating relationships among these features.

## 4 EXPERIMENTS

We conduct experiments on both simulation and real HSI datasets. Following the approaches in Meng et al. (2020b;a); Huang et al. (2021); Cai et al. (2022b), we select a set of 28 wavelengths ranging from 450-650nm by employing spectral interpolation techniques applied to the HSI data.

### 4.1 EXPERIMENTAL SETTINGS

**Simulation and Real Datasets:** We adopt two widely used HSI datasets, i.e., CAVE Park et al. (2007) and KAIST Choi et al. (2017) for simulation experiments. The CAVE dataset comprises 32 HSIs with a spatial size of $512 \times 512$. The KAIST dataset includes 30 HSIs with a spatial size of $2704 \times 3376$. Following previous works Meng et al. (2020b;a); Huang et al. (2021); Cai et al. (2022b), we employ the CAVE dataset as the training set, while 10 scenes from the KAIST dataset are utilized for testing. During the training process, a real mask of size $256 \times 256$ pixles is applied. In our real experiment, we utilized the HSI dataset captured by the SD-CASSI system in Meng et al. (2020b). The system captures real-world scenes of size $660 \times 714 \times 28$ with wavelengths spanning from 450 to 650 nm and dispersion of 54 pixels.

**Implementation Details:** For the codebook settings, the item number $N$ of the codebook is set to 4096, and the code dimension $d$ is set to 112. In the first phase of training, the trade-off weight $\beta = 0.25$. For all phases of training, we use the Adam Kingma & Ba (2014) optimizer with a batch size of 4. and set the learning rate to $4 \times 10^{-4}$. PSNR and SSIM Wang et al. (2004) are utilized as our metrics. Our method is implemented with the PyTorch framework and trained using four NVIDIA RTX3090 GPUs.

Table 1: The average results of PSNR in dB (top entry in each cell), SSIM (bottom entry in each cell) on the 10 synthetic spectral scenes.

| Algorithms | Scene1 | Scene2 | Scene3 | Scene4 | Scene5 | Scene6 | Scene7 | Scene8 | Scene9 | Scene10 | Avg |
|---|---|---|---|---|---|---|---|---|---|---|---|
| TwIST | 25.16 | 23.02 | 21.40 | 30.19 | 21.41 | 20.95 | 22.20 | 21.82 | 22.42 | 22.67 | 23.12 |
| | 0.700 | 0.604 | 0.711 | 0.851 | 0.635 | 0.644 | 0.643 | 0.650 | 0.690 | 0.569 | 0.669 |
| GAP-TV | 26.82 | 22.89 | 26.31 | 30.65 | 23.64 | 21.85 | 23.76 | 21.98 | 22.63 | 23.10 | 24.36 |
| | 0.754 | 0.610 | 0.802 | 0.852 | 0.703 | 0.663 | 0.688 | 0.655 | 0.682 | 0.584 | 0.669 |
| DeSCI | 27.13 | 23.04 | 26.62 | 34.96 | 23.94 | 22.38 | 24.45 | 22.03 | 24.56 | 23.59 | 25.27 |
| | 0.748 | 0.620 | 0.818 | 0.897 | 0.706 | 0.683 | 0.743 | 0.673 | 0.732 | 0.587 | 0.721 |
| HSSP | 31.48 | 31.09 | 28.96 | 34.56 | 28.53 | 30.83 | 28.71 | 30.09 | 30.43 | 28.78 | 30.35 |
| | 0.858 | 0.842 | 0.823 | 0.902 | 0.808 | 0.877 | 0.824 | 0.881 | 0.868 | 0.842 | 0.852 |
| DNU | 31.72 | 31.13 | 29.99 | 35.34 | 29.03 | 30.87 | 28.99 | 30.13 | 31.03 | 29.14 | 30.74 |
| | 0.863 | 0.846 | 0.845 | 0.908 | 0.833 | 0.887 | 0.839 | 0.885 | 0.876 | 0.849 | 0.863 |
| DGSMP | 33.26 | 32.09 | 33.06 | 40.54 | 28.86 | 33.08 | 30.74 | 31.55 | 31.66 | 31.44 | 32.63 |
| | 0.915 | 0.898 | 0.925 | 0.964 | 0.882 | 0.937 | 0.886 | 0.923 | 0.911 | 0.925 | 0.917 |
| HDNet | 35.14 | 35.67 | 36.03 | 42.30 | 32.69 | 34.46 | 33.67 | 32.48 | 34.89 | 32.38 | 34.97 |
| | 0.935 | 0.940 | 0.943 | 0.969 | 0.946 | 0.952 | 0.926 | 0.941 | 0.942 | 0.937 | 0.943 |
| MST++ | 35.40 | 35.87 | 36.51 | 42.27 | 32.77 | 34.80 | 33.66 | 32.67 | 35.39 | 32.50 | 35.99 |
| | 0.941 | 0.944 | 0.953 | 0.973 | 0.947 | 0.955 | 0.925 | 0.948 | 0.949 | 0.941 | 0.951 |
| CST-L+ | 35.96 | 36.84 | 38.16 | 42.44 | 33.25 | 35.72 | 34.86 | 34.34 | 36.51 | 33.09 | 36.12 |
| | 0.949 | 0.955 | 0.962 | 0.975 | 0.955 | 0.963 | 0.944 | 0.961 | 0.957 | 0.945 | 0.957 |
| DAUHST-9stg | 37.25 | 39.02 | 41.05 | 46.15 | 35.80 | 37.08 | 37.57 | 35.10 | 40.02 | 34.59 | 38.36 |
| | 0.958 | 0.967 | 0.971 | 0.983 | 0.969 | 0.970 | 0.963 | 0.966 | 0.970 | 0.956 | 0.967 |
| DADF-Net (Plus-3) | 37.46 | 39.86 | 41.03 | 45.98 | 35.53 | 37.02 | 36.76 | 34.78 | 40.07 | 34.39 | 38.29 |
| | 0.965 | 0.976 | 0.974 | 0.989 | 0.972 | 0.975 | 0.958 | 0.971 | 0.976 | 0.962 | 0.972 |
| RDLUF-MixS2-3stg | 36.67 | 38.48 | 40.63 | 46.04 | 34.63 | 36.18 | 35.85 | 34.37 | 38.98 | 33.73 | 37.56 |
| | 0.953 | 0.965 | 0.971 | 0.986 | 0.963 | 0.966 | 0.951 | 0.963 | 0.966 | 0.950 | 0.963 |
| Ours-3stg | 36.86 | 38.46 | 40.83 | 45.92 | 35.01 | 36.43 | 36.41 | 34.90 | 39.31 | 33.80 | 37.90 |
| | 0.959 | 0.967 | 0.974 | 0.988 | 0.967 | 0.971 | 0.958 | 0.968 | 0.971 | 0.956 | 0.969 |
| RDLUF-MixS2-9stg | 37.94 | 40.95 | 43.25 | 47.83 | 37.11 | 37.47 | 38.58 | 35.50 | 41.83 | 35.23 | 39.57 |
| | 0.966 | 0.977 | 0.979 | 0.990 | 0.976 | 0.975 | 0.969 | 0.970 | 0.978 | 0.962 | 0.974 |
| Ours-9stg | 37.98 | 41.20 | 43.34 | 47.70 | 37.16 | 37.82 | 38.61 | 36.44 | 42.64 | 35.29 | 39.82 |
| | 0.969 | 0.981 | 0.979 | 0.992 | 0.977 | 0.978 | 0.970 | 0.978 | 0.983 | 0.967 | 0.977 |

Figure 4: The synthetic data comparisons. 4 out of 28 wavelengths are selected to compare visually. 'Corr' is the correlation coefficient between one method curve and the ground truth curve of the chosen (green box) region. Our method has a more accurate wavelength curve than others.

## 4.2 COMPARE WITH STATE-OF-THE-ART

We compare our proposed method with recent state-of-the-art (SOTA) methods including deep unfolding series RDLUF-MixS2 Dong et al. (2023), DAUHST Cai et al. (2022c), end-to-end designed networks DADF-Net Xu et al. (2023), CST Cai et al. (2022a), MST Cai et al. (2022b), HDNetHu et al. (2022) and traditional model-based methods TwIST Bioucas-Dias & Figueiredo (2007) and DeSCI Liu et al. (2019) on both synthetic and real datasets. Other methods like GAP-TV Yuan (2016), HSSP Wang et al. (2019), DNU Wang et al. (2020) and DGSMP Huang et al. (2021) are compared on synthetic data.

**Synthetic data:** Table 1 shows quantitative comparisons on synthetic data that our proposed method outperforms other compared methods. It surpasses the recent deep unfolding method RDLUF-MixS2 both in the 3-stage (+0.34dB) and 9-stage (+0.25 dB) network according to average PSNR and SSIM. The Fig. 4 shows the visual reconstruction results. 4 out of 28 wavelengths are selected

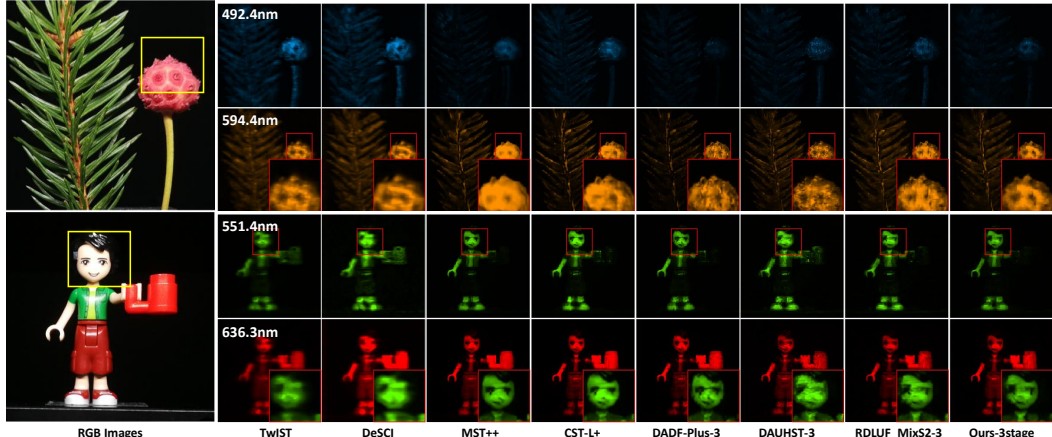

Figure 5: The real data comparisons of 2 scenes. 4 out of 28 wavelengths are selected to compare visually.

to compare. The region within the red box was chosen to analyze the wavelength accuracy. 'Corr' is the correlation coefficient between one method and the ground truth of the chosen region. According to the 'Corr', our method (0.99) has a more accurate wavelength curve than others (0.98 for RDLUF-MixS2). Moreover, as shown in Fig. 1, our method demonstrates a stronger capability to reconstruct intricate details and textures, while previous methods tend to over-smooth the surface. The visualization of the learned knowledge in (b) further illustrates the learned patterns facilitate improved image fidelity. Each patch in (b) denotes a codebook value which is decoded to the natural image domain to represent some basic texture elements.

**Real data:** Fig. 5 shows reconstruction results of multiple methods on the real scene 'Lego Man' and 'Real Plant'. 4 out of 28 wavelengths are selected to compare visually. Unlike the previous SOTA RDLUF-MixS2, we can see that our method can reconstruct fewer artifacts horizontally and vertically. More results can be seen in the supplementary material (SM).

## 5 ABLATION STUDY

For the ablation study, we train our model on the synthetic training data with 3 unfolding stage models. Table 5 shows our contributions yield quality performance improvements. We only show main module comparisons here, further ablation study can also be seen in the SM. Row 4 in Ta-

| Method | PSNR (dB) | Params (M) |
|---|---|---|
| w/o MCE and MF | 37.61 | 2.317 |
| w/o HFK in MKA | 37.74 | 2.322 |
| plain ADMM projection | 37.45 | 2.187 |
| w/o two-phase strategy | 37.80 | 2.430 |
| Our Full Model | 37.90 | 2.325 |

Figure 6: Ablation study of our method.

ble 5 means that it only uses normal denoiser in the first stage unfolding and only trains the network once from measure to reconstruction. $\mathbf{F}_q$ is also removed in MKA. It demonstrates that our two-phase learning strategy leads to a 0.1 dB improvement. Row 1 in Table 5 removes mask encoder and $\mathbf{F}_M^i$ in attention. It demonstrates that our mask cross encoder with cross attention improves the performance by 0.29 dB. Row 2 uses two-phase training but $\mathbf{F}_q$ in attention. It demonstrates that our prior knowledge of cross attention improves the performance by 0.16 dB. Row 3 in Table 5 uses a plain ADMM projection in our unfolding network, dropping the performance by 0.45 dB. It illustrates that our DE-ADMM plays a role in the unfolding network.

## 6 CONCLUSION

This paper introduces a spectral SCI reconstruction network that leverages high-fidelity knowledge from clean HSIs as well as the cascade degradation information. It achieves state-of-the-art performance on simulated data and real data, surpassing the previous best reconstruction algorithms both in spatial and spectral dimensions. In addition, we proposed a two-phase training strategy within a deep unfolding framework to transfer essential knowledge. We hope to give a new way of modeling high-fidelity knowledge from ground-truth observation space for video compressive, which can potentially extend to a wider range of application scenarios in the future, such as video SCI.

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
