# OpenReview forum: "Degradation-aware Unfolding Knowledge-assist Transformer for Spectral Compressive Imaging"
_ICLR.cc/2024/Conference — ICLR 2024 Conference Withdrawn Submission_

### Official Review · Reviewer_y4To · 2023-10-30

**Soundness:** 3 good
**Presentation:** 3 good
**Contribution:** 2 fair
**Rating:** 5
**Confidence:** 4

**Summary:**

The paper introduces a degradation-aware unfolding knowledge-assist transformer for spectral compressive imaging. It aims to address the challenge of reconstructing high-fidelity 3D spectral signals from 2D measurements. The authors propose a two-phase training strategy that embeds high-quality knowledge prior in a deep unfolding framework. They employ a lightweight counselor model trained via a vector-quantized variational autoencoder (VQ-VAE) to guide the reconstruction process. Experiments are conducted to evaluate the performance of  the method against existing ones.

**Strengths:**

1. The presentation of the paper is satisfactory, and the main idea is easy to access
2. The experiments showed modest improvement over existing ones.

**Weaknesses:**

1. This  paper is closely related to [2022c] Cai et al. "Degradation-aware unfolding half-shuffle transformer for spectral compressive imaging, NeurIPS2022. Its main idea essentially is similar to that of [2022c]. The paper did not provide a detailed comparison on the main differences and why the proposed one works better than [2022c].

2, The work appears to be a re-engineered version of existing research, particularly the aforementioned work by Cai et al. (2022). The paper lacks compelling arguments to substantiate the purported benefits of the modifications introduced in their method.

3. Given the application-oriented nature of this research, the limited dataset employed for testing undermines the credibility of the claimed performance gains over existing methods.

4. While the ablation studies do indicate minor improvements attributable to the proposed method, these gains are not substantial enough to fully support the paper's claims. It raises the question of whether the observed performance improvements might be more a result of fine-tuning the neural network rather than the novel aspects of the proposed method.

**Questions:**

See the discussion above

---

### Official Review · Reviewer_EFVD · 2023-10-31

**Soundness:** 3 good
**Presentation:** 3 good
**Contribution:** 2 fair
**Rating:** 3
**Confidence:** 5

**Summary:**

This paper proposes a degradation-aware knowledge-assist deep unfolding framework for preserving fine-grained details. The denoiser is learned with a VQ-VAE-based counselor network and a sophisticated mask cross encoder. The proposed method achieves comparable experimental results on synthetic benchmarks and real-world datasets

**Strengths:**

1.	The proposed method achieves the SOTA results.
2.	The overall paper is easy to follow.

**Weaknesses:**

1.	The contribution of this paper is incremental. The main contribution is combining a VQ-VAE in the first stage of the Deep Unfolding framework. Although the idea of utilizing a codebook makes sense, the current approach is very trivial.
2.	Visual performance is poor. In Figure 4, the result of RDLUF_MixS2-9 is clearer in details than result of the proposed method. Moreover, Moreover, CST-L+ seems to obtain better visual effects than the proposed method in real experiments. This is inconsistent with the author's claim that more texture information can be restored
3.	Lack of sufficient experimental verification. The number and location of the VQ-VAE module should have an impact on the results, and the author should provide relevant experiments
4.	Lack of comparison of computational cost and parameter numbers to prove the proposed method is more efficient. The complexity of Deep Unfolding methods is not only reflected in the number of stages, but also in the computational complexity of each stage

**Questions:**

1.	In the two-stage training of the network, there is a significant difference in the input of VQ-VAE. The first stage is a clean image, and the second stage is a very degraded image. Why not put VQ-VAE at the end of the network to make its input more similar to the input of the first stage? Would it be better to replace all denoisers in the network with VQ-VAE?
2.	What is the meaning of ‘video’ in the sentence ‘with a counselor model from the ground truth video as external knowledge prior and aim to guide the reconstruction’ in the 2.2 section?

---

### Official Review · Reviewer_fgj2 · 2023-10-31

**Soundness:** 2 fair
**Presentation:** 2 fair
**Contribution:** 2 fair
**Rating:** 5
**Confidence:** 4

**Summary:**

In this paper, a spectral SCI reconstruction network was introduced, which incorporated high-fidelity knowledge from clean HSIs and explicit cascade degradation information into a deep unfolding framework for spectral compressive image reconstruction. A two-phase training strategy was employed, starting with a lightweight counselor model based on a vector quantized variational autoencoder (VQ-VAE) to learn high-fidelity knowledge from ground-truth images. Subsequently, a U-net bridge denoiser equipped with the mask-aware knowledge-assist attention (MKA) mechanism was introduced to incorporate multi-layer degradation information and high-fidelity prior knowledge effectively. Importantly, the unfolding framework could integrate external prior knowledge from the counselor model and cascade degradation information, enhancing the reconstruction performance. The key contributions included the proposal of a degradation-aware knowledge-assist deep unfolding framework, a U-net bridge denoiser that combined high-fidelity HSI knowledge and cascade degradation information, and extensive experimental validation demonstrating the method's superior accuracy in both spatial and spectral dimensions, using synthetic benchmarks and real datasets.

**Strengths:**

The paper contributes to the field of image processing by addressing challenges related to degradation mismatch and offering a framework that can enhance the quality of spectral compressive image reconstructions.

**Weaknesses:**

The ablation experiments are conducted exclusively on synthetic data. To enhance the comprehensiveness of the research, it would be beneficial to include experiments on real datasets as well, offering a more comprehensive evaluation of the proposed method's effectiveness.

The terminology "bridge U-net" appears to refer to a U-net framework with transformer embeddings, and it seems that the bridges between the encoder and decoder functions are just skip connections. Providing clarity on these terms and their roles within the model would aid in understanding the architectural components.

**Questions:**

A significant challenge addressed in the paper is the impact of degradation mismatch on reconstructed images. It might be worthwhile to explore the possibility of applying sophisticated denoising techniques as an initial step before proceeding with other processing stages. This approach could potentially lead to improved results and mitigate the issues arising from degradation mismatch.

The authors mention that the first stage denoiser is a pre-trained VQ-VAE, while subsequent stages use U-shape networks. It is essential to justify the necessity of designing a special module for the first stage and explain the criteria used for selecting the type and size of these modules in the following stages. Providing these justifications would enhance the transparency and rationale behind the model's design.

---

### Official Review · Reviewer_wS7P · 2023-11-01

**Soundness:** 3 good
**Presentation:** 2 fair
**Contribution:** 2 fair
**Rating:** 5
**Confidence:** 4

**Summary:**

This paper proposed a degradation-aware deep unfolding neural network for spectral compressive imaging. The unfolding neural network is assisted by the vector quantized hyperspectral image knowledge, integreated with the degradation information via a U-net bridge denoiser. A  two-phase scheme is introduced for model traiing. Extensive experiments on the synthetic benchmark wereconducted demonstrate the effectiveness of theproposed method.

**Strengths:**

1. Superior reconstruction accuracy has been achieved over existing works, in both spatial and spectral dimension.

2. The proposed network is degradation-aware and knowledge-assisted.

3. Utilization of convolutions, transformer blocks and maks-aware self-attention in an unfolding network leads to performance gain.

**Weaknesses:**

1. The novelty of this approach appears limited. Unfolding networks with transformers or self-attention mechanisms have been previously introduced in previous works, such as [A1] for spectral compressive imaging. Additionally, the U-net bridge denoiser can be described as skip connections augmented with multiple convolution layers (as mentioned on Page 7, "we implement a bridge module including several convolution layers"), which is a common architectural element found in existing literature.

[A1] Degradation-Aware Unfolding Half-Shuffle Transformer for Spectral Compressive Imaging NIPS 2022.


2. The MASK-AWARE KNOWLEDGE-ASSIST DENOISING moduel integrates mask priors into the Key and Value components of Self-Attention. This approach shares similarities with the one proposed in [A2]. Furthermore, the experimental results provide support for the advantages of incorporating mask priors into the Values of Self-Attention.

[A2] Mask-guided Spectral-wise Transformer for Efficient Hyperspectral Image Reconstruction, CVPR 2022.

3. The experimental part misses the comparison in terms of computational compleixty, specifically the number of FLOPS and running time. Indeed, the proposed model appears to be computationally intensive, due to its utilization of deformable and self-attention operations.


4. The performance improvement achieved through the two-stage scheme is marginal, with only a 0.1dB PSNR gain.

5. The writing needs improvement. For instance, the Abstrtact mentions little information of the proposed method.

6. The zoom-in regions seem to be mistakenly into the last rows, in both Fig. 4 and Fig. 5.

**Questions:**

1. Please clarify the difference between this work and [A1] as well as [A2].

2. Please compare in terms of the number of FLOPS and running time.